# BertNet: Harvesting Knowledge Graphs from Pretrained Language Models

## Abstract

Symbolic knowledge graphs (KGs) have been constructed either by expensive human crowdsourcing or with complex text mining pipelines. The emerging large pretrained language models (LMs), such as BERT, have shown to implicitly encode massive knowledge which can be queried with properly designed prompts. However, compared to the explicit KGs, the implicit knowledge in the black-box LMs is often difficult to access or edit and lacks explainability. In this work, we aim at harvesting symbolic KGs from the LMs, and propose a new framework for automatic KG construction empowered by the neural LMs' flexibility and scalability. Compared to prior works that often rely on large human annotated data or existing massive KGs, our approach requires only the minimal definition of relations as inputs, and hence is suitable for extracting knowledge of rich *new* relations that are instantly assigned and not available before. The framework automatically generates diverse prompts, and performs efficient knowledge search within a given LM for consistent outputs. The knowledge harvested with our approach shows competitive quality, diversity, and novelty. As a result, we derive from diverse LMs a family of new KGs (e.g., BertNet and RobertaNet) that contain a richer set of relations, including some complex ones (e.g., `"A is capable of but not good at B"`) that cannot be extracted with previous methods. Besides, the resulting KGs also serve as a vehicle to interpret the respective source LMs, leading to new insights into the varying knowledge capability of different LMs.

## 1 Introduction

Symbolic knowledge graphs (KGs) encode rich knowledge about entities and their relationships, and have been one of the major means for organizing commonsense or domain-specific information to empower various applications, including search engines (Xiong et al., 2017; Google, 2012), recommendation systems (Wang et al., 2019a; 2018; 2019b), chatbots (Moon et al., 2019; Liu et al., 2019b), healthcare (Li et al., 2019; Mohamed et al., 2020; Lin et al., 2020), etc. The common practice for constructing a KG is crowdsourcing (such as ConceptNet (Speer et al., 2017), WordNet (Fellbaum, 2000), and ATOMIC (Sap et al., 2019)) , which is accurate but often has limited coverage due to the extreme cost of manual annotation (e.g., ConceptNet covers only 34 types of commonsense relations). Prior work has also built text mining pipelines to automatically extract knowledge from unstructured text, including domain-specific knowledge (Wang et al., 2021b) and commonsense knowledge (Zhang et al., 2020; Romero et al., 2019; Nguyen et al., 2021). Those systems, however, often involve a complex set of components (e.g., entity recognition, coreference resolution, relation extraction, etc.), and applicable only to a subset of all the knowledge, which is explicitly stated in the text.

On the other hand, the emerging large language models (LMs) pretrained on massive text corpora, such as BERT (Devlin et al., 2019), RoBERTa (Liu et al., 2019a), and GPT-3 (Brown et al., 2020), have been shown to encode a large amount of knowledge implicitly in their parameters. This has inspired the interest in using the LMs as knowledge bases. For example, recent work has focused on manually or automatically crafted prompts (e.g., `"Obama was born in ___"`) to query the LMs for answers (e.g., `"Hawaii"`) (Petroni et al., 2019; Jiang et al., 2020; Shin et al., 2020; Zhong et al., 2021) . Such probing also serves as a way to interpret the black-box LMs (Swamy et al., 2021), and inspires further fine-tuning to improve knowledge quality (**?**Newman et al., 2021; Fichtel et al., 2021). However, the black-box LMs, where knowledge is only implicitly encoded, fall short of the many nice properties of explicit KGs (AlKhamissi et al., 2022), such as the easiness of browsing

| Method | Knowledge source | Outcome | Arbitrary relation |
|--------|-----------------|---------|---------------------|
| Text mining | text corpus | KG | ✗ |
| Factual probing | LMs | tail entity | ✓ |
| COMET | LMs + Existing Knowledge | tail entity | ✗ |
| Symbolic KD | GPT-3 (+ existing KGs) | KG | ✓* |
| BertNet (**Ours**) | LMs | KG | ✓ |

Table 1: Categorization of works on automated knowledge graph construction. Compared with others, our framework is more flexible as it relies on **LMs** as knowledge source, generates a full **KG**, applies to **arbitrary relations**. The details are presented in Section 2.

the knowledge or even making updates (Zhu et al., 2020; Cao et al., 2021) and the explainability for trustworthy use by domain experts. Can we automatically *harvest KGs from the LMs*, and hence combine the best of both worlds, namely the flexibility and scalability from the neural LMs, and the access, editability, and explainability in the symbolic form?

This paper makes a step towards this end—given a pretrained LM (e.g., BERT, ROBERTA), our automatic framework extracts a KG from it in an efficient and scalable way—this results in a family of new KGs (referred to as, e.g., BERTNET, ROBERTANET) that permit a new *extendable* set of knowledge beyond the existing hand-annotated KGs such as ConceptNet. As Table 1 suggests, our framework is more flexible compared with previous works to construct knowledge graphs automatically.

The new goal of extracting KGs purely from any LMs poses unique challenges. First, it is difficult to extract knowledge reliably from LMs, as the LMs have shown to generate inconsistent outputs given prompts of slightly different wording (e.g., "___ originally aired in ___" vs "___ premiered on ___") (Elazar et al., 2021; Newman et al., 2021). Automatically learning the optimal prompts (Lester et al., 2021; Zhong et al., 2021; Qin & Eisner, 2021) typically requires many existing entity pairs as training data, which is often not available especially for *new* relations that are instantly assigned. While West et al. (2021) extracted commonsense knowledge of high quality from GPT-3, this method doesn't apply to other LMs, since it relies on the extreme few-shot learning ability and the large capacity of GPT-3 model. To this end, we apply an unsupervised method that automatically paraphrases an initial prompt and create a diverse set of alternative prompts with varying confidence weights. We then search for entity pairs that consistently satisfy the diverse prompts. The second challenge lies in the search phase due to the large space of entity (one or multiple tokens) tuples. We devise an efficient search-and-rescoring strategy that strikes the balance between knowledge accuracy and coverage.

The minimal dependence on other sources besides the powerful LM itself allows maximal flexibility of our framework to extract novel knowledge, such as those about complex relations like "A is capable of, but not good at, B" that expresses sophisticated meaning and "A can do B at C" that involves multiple entities. Besides, the resulting KGs can readily serve as a symbolic interpretation of the respective black-box LMs, for users to browse and understand their knowledge storage and capability.

We apply the framework to harvest KGs from a wide range of popular LMs, including ROBERTA, BERT, and DISTILBERT, of varying model sizes, respectively. The experiments show our approach can harvest large-scale KGs of diverse concepts, and performs well on user-defined complex relations.

Compared with other KGs trained with existing knowledge bases or human annotations, the outcome KGs of our framework, with solely a LM as the knowledge source, shows competitive quality, diversity and novelty. The further analysis illustrates the better balance of knowledge accuracy and coverage than baselines. Comparison between the resulting KGs from different LMs offers new insights into their knowledge capacities due to different factors, such as model sizes, pretraining strategies, and distillation.

## 2 RELATED WORK

**Knowledge graph construction**  Popular knowledge bases or KGs are usually constructed with heavy human labor. For example, WordNet (Fellbaum, 2000) is a lexical database that links words into semantic relations; ConceptNet (Speer et al., 2017) is a large commonsense knowledge graph in the conventional KG format that consists of (head entity, relation, tail entity) triples; ATOMIC (Sap et al., 2019) is a social commonsense KG by crowdsourcing consisting of if-then statements. Automatic Knowledge Graph Construction has been focused by academics. We summarize different categories of works in Table 1. Text mining based works aim to extract knowledge from text. A typical information extraction system (Angeli et al., 2015) decomposes the task to a set of sub-tasks, such as co-reference resolution, named entity recognition, and relationship extraction. There are also works on commonsense knowledge extraction, like WebChild (Tandon et al., 2014), TransOMCS (Zhang et al., 2020), DISCOS (Fang et al., 2021b), Quasimod o(Romero et al., 2019), ASCENT (Nguyen et al., 2021). These extraction pipelines are based on linguistic pattern, and involve complex engineering such as corpus selection, term aggregation, filtering and so on. Recently there are attempts to make use of language models. Wang et al. (2021a) finetuned LMs to predict missed links in KGs. COMET (Bosselut et al., 2019) is a finetuned generative LM trained to generate tail entities given head entities and relations. Symbolic knowledge distillation (KD) (West et al., 2021) distil the knowledge in GPT-3 to a generative LM. As a intermediate product, they got $ATOMIC_{10x}$ by prompting GPT-3 (Brown et al., 2020) with examples. However, this method only applies to GPT-3 for the need of strong few-shot learning ability. Factual probing (Petroni et al., 2019; AlKhamissi et al., 2022; Razniewski et al., 2021) measures the amount of knowledge in LMs by the accuracy when filling a blank in a prompt, which is human-written, from text mining, or learned with a large amount of existing knowledge. We summarize their difference to our work in the caption of Table 1 To the best of our knowledge , our framework is the first to construct a knowledge graph via extracting knowledge purely from LMs (with minimal definition of relations as input).

**LMs as knowledge bases**  It is easy to query knowledge in knowledge bases, while the implicit knowledge in LMs are difficult to access. The retrieval of knowledge in LMs often requires finetuning (Da et al., 2021; Wallat et al., 2020) or prompt tuning (Qin & Eisner, 2021; Jiang et al., 2020; Adolphs et al., 2021; Davison et al., 2019; Petroni et al., 2019) with existing knowledge, limiting their methods to query new relations without existing tuples. Together with the success of pretrained LMs , recent work has developed various ways to understand their internal mechanisms, such as analyzing the internal states of LMs, and extracting structured linguistic patterns (Tenney et al., 2019; Hewitt & Manning, 2019). **Factual probing** aims to quantify the factual knowledge in pretrained language models, which is usually implemented by prompting methods and leveraging the masked language model pretraining task. Specifically, the amount of factual knowledge is estimated by a set of human-written cloze-style prompts, e.g., `"Dante was born in ___"`. The accuracy of the model prediction on the blank represents a lower bound of the amount of knowledge in the model. LAMA (Petroni et al., 2019) collects a set of human-written prompts to detect the amount of factual information that a masked language model encodes. LPAQA (Jiang et al., 2020) proposes to use text mining and paraphrasing to find and select prompts to optimize the prediction of a single or a few correct tail entities, instead of extensively predicting all the valid entity pairs like in our framework. AutoPrompt (Shin et al., 2020) and OPTIPrompt (Zhong et al., 2021) search prompts automatically. Though making prompts unreadable, they achieve higher accuracy on the knowledge probing tasks. Instead of measuring knowledge inside an LM as an accuracy number, our framework explicitly harvests a KG from the LM. **Consistency** is a significant challenge in knowledge probing and extraction, which refers to a model that should not have predictions contradicting each other. Basically, models should behave invariantly under inputs with different surface forms but the same meaning, e.g., paraphrased sentences or prompts. Several benchmarks are proposed to study consistency in LMs (Elazar et al., 2021; Hase et al., 2021). Elazar et al. (2021) analyzes the consistency of pretrained language models with respect to the factual knowledge. They show that the consistency of all LMs is poor in their experiment. In our framework, the extracted entity pairs for each relation would consistently satisfy a diverse set of prompts. Unlike traditional KGs, it is also non-trivial to edit knowledge in LMs, though there is some preliminary research on editing facts in LMs (Zhu et al., 2020; Cao et al., 2021), via forcing the model to change predictions on some data points while keeping the same for other data points. There is also another line of works that utilize LMs to score entity pairs (Feldman et al., 2019; Bouraoui et al., 2020; Fang et al., 2021b;a) . However, these works only use LMs to filter candidate knowledge tuples that are collected either

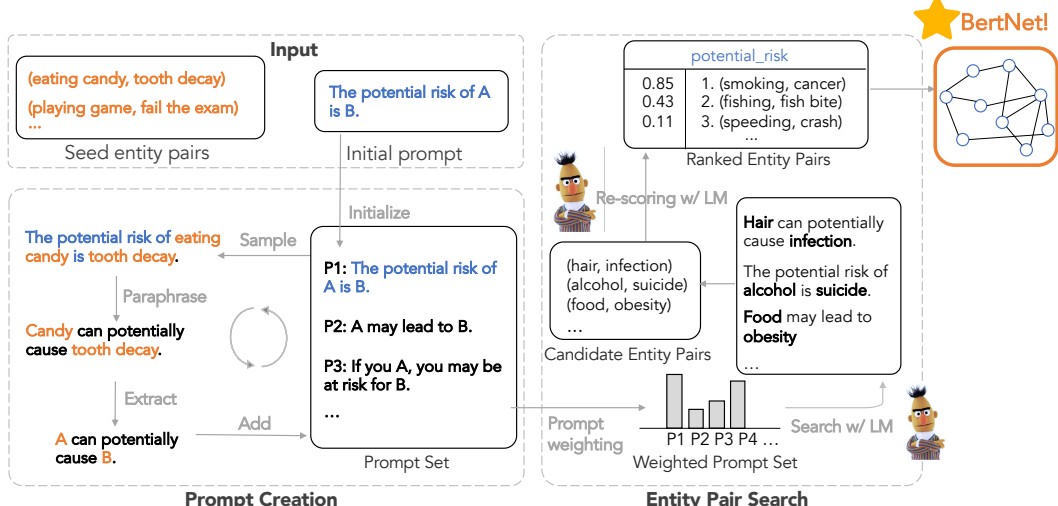

Figure 1: An overview of the knowledge harvesting framework. Given the minimal definition of the relation as input (an initial prompt and a few shot of seed entity pairs), the approach first automatically creates a set of prompts expressing the relation in a diverse ways (§3.1). The prompts are weighted with confidence scores. We then use the LM to search a large collection of candidate entity pairs, followed by re-scoring/ranking that yields the top entity pairs as the output knowledge (§3.2).

from the Internet or existing KGs, but they don't direclty extract knowledge inside the LMs as our framework.

## 3 HARVESTING KGS FROM LMS

This section presents the proposed framework for extracting a relational KG from a given pretrained LM, where the LM can be arbitrary fill-in-the-blank models such as BERT (Devlin et al., 2019), ROBERTA (Liu et al., 2019a), BART (Lewis et al., 2020), or GPT-3 (with appropriate instructions) (Brown et al., 2020). The KG consists of a set of knowledge tuples in the form `<head entity (h), relation (r), tail entity (t)>`. Specifically, we automatically harvest from the LM all appropriate entity pairs $(h, t)$ for any given relation $r$. This presents a substantially more challenging problem than the popular LM probing tasks (Petroni et al., 2019; Jiang et al., 2020; Shin et al., 2020; Zhong et al., 2021) which predict a particular tail entity given both the head entity and relation. In contrast, given only the relation, we need to search for hundreds or thousands of head/tail entity pairs both accurately and extensively in an efficient manner.

To extract knowledge tuples of a particular relation (e.g., `"potential_risk"` as illustrated in Figure 1), our approach requires only minimal input information that defines the relation of interest, namely an initial prompt (e.g., `The potential risk of A is B`) together with a few shot of seed entity pairs (e.g., `<eating candy, tooth decay>`). The prompt offers the overall semantics of the relation, while the seed entity pairs serve to eliminate any possible ambiguities. For new relations not included in existing KGs, it is impractical to require a large set (e.g., hundreds) of seed entity pairs as in previous knowledge probing or prompt optimization methods (Petroni et al., 2019; Jiang et al., 2020; Shi et al., 2019; Zhong et al., 2021). Instead, our approach needs only a very small set, such as 5 example pairs in our experiments, which can easily be collected or written by users.

A key challenge of directly feeding the initial prompt and asking the LM to generate head/tail entities is that the outputs are often inconsistent: for example, slightly changing the wording of the prompt (while keeping the same semantic) can easily lead to different irrelevant LM outputs (Elazar et al., 2021; Hase et al., 2021). The inconsistency renders the extracted knowledge unreliable and often inaccurate (as also shown in our experiments). Inspired by and generalizing the prior knowledge probing work (Petroni et al., 2019; Jiang et al., 2020; Shin et al., 2020; Zhong et al., 2021), we overcome the challenge by automatically creating diverse weighted prompts describing the same relation of interest, and searching for those entity pairs that are consistently favored by the

multiple prompts under the LM. Interestingly, we found that those automatically created prompts, though stemming from the initial prompt (often written by a human), are often better alternatives than the initial prompt itself for drawing more accurate knowledge from the LMs, as studied in our experiments.

In the following sections, we describe the core components of our approach, namely the automatic creation of diverse prompts with confidence weights (§3.1) and the efficient search to discover consistent entity pairs (§3.2) that compose the desired KGs. Figure 1 illustrate the overall framework.

## 3.1 AUTOMATIC CREATION OF DIVERSE WEIGHTED PROMPTS

Given the input information of a relation, namely the initial prompt and a few shots of seed entity pairs, we automatically paraphrase the initial prompt to a large set of prompts that are linguistically diverse but semantically describe the same relation. Each of the prompts is further associated with a confidence weight for an accurate measure of knowledge consistency in the next section.

Specifically, starting with the initial prompt, we randomly sample an entity pair from the seed set and insert it into the prompt to form a complete sentence. We then use an off-the-shelf text paraphrase model (see §4.1 for more details) to produce multiple paraphrased sentences of the same meaning. By removing the entity names, each paraphrased sentence results in a new prompt that describes the desired relation. To ensure diverse expressions of the relation, we keep those prompts that are sufficiently different from each other in terms of edit distance. We repeat this process by continuing the paraphrasing of the newly created prompts, until we collect at least 10 prompts for the relation.

The automatic creation of prompts is inevitably noisy. Some of the resulting prompts are less precise in expressing the exact meaning of the desired relation. we define a reweighting score for each prompt to calibrate its effect in the next knowledge search step. Specifically, we measure the compatibility of the new prompts with the seed entity pairs, and derive the prompt weights with the compatibility scores. Intuitively, given an entity pair $\langle h, t \rangle$, a compatible prompt $p$ should induce a high likelihood under the LM for each single entity plugged into the prompt. That is, the minimum likelihood of $h$ and $t$ should be high. Besides, we also expect a high joint likelihood of $h$ and $t$. Formally, the compatibility score is written as:

$$f_{LM}(\langle h, t \rangle, p) = \alpha \log P_{LM}(h, t|p) + (1 - \alpha) \min \{\log P_{LM}(h|p), \log P_{LM}(t|p, h)\} \quad (1)$$

where the first term is the joint log-likelihood under the LM distribution $P_{LM}$, the second term is the minimum individual log-likelihood given the prompt (and the other entity), and $\alpha$ is a balancing factor for which we set $\alpha = 2/3$ in our experiments.

We compute the average compatibility score of each created prompt over all seed entity pairs. The weight of the prompt is then defined as the softmax-normalized score across all prompts.

## 3.2 EFFICIENT SEARCH FOR CONSISTENT KNOWLEDGE TUPLES

Given the above set of prompts with confidence weights, we next harvest entity pairs that consistently satisfy all the prompts. Each entity consists of one or more tokens. We reuse the above prompt/entity-pair compatibility function (Eq.1) and intuitively define the *consistency* of a new entity pair $\langle h^{new}, t^{new} \rangle$ as the weighted average of its compatibility with the different prompts:

$$\text{consistency}(\langle h^{new}, t^{new} \rangle) = \sum_p w_p \cdot f_{LM}(\langle h^{new}, t^{new} \rangle, p) \quad (2)$$

where $w_p$ is the prompt weight and the sum is over all automatically created prompts as above. Therefore, an entity pair that is compatible with all prompts are considered to be consistent.

Based on the consistency criterion, we devise an efficient search strategy for consistent entity pairs. A straightforward approach is to simply enumerate all possible pairs of entities, measure the respective consistency scores as above, and pick the top-K entity pairs with highest scores as the resulting knowledge for the KG. The approach can be prohibitively slow given the large vocabulary size $V$ and the high time complexity of the enumeration (e.g., $O(V^2)$ assuming each entity consists of only one token).

To this end, we an appropriate approximation that leads to a more efficient *search and re-scoring* method. More concretely, we first use the minimum individual log-likelihoods (namely, the second

term in the compatibility score Eq.1), weighted averaged across different prompts (similar as in Eq.2), to propose a large set of candidate entity pairs. The use of the minimum individual log-likelihoods allows us to apply a pruning strategy, as described more in the appendix. Once we collect a large number of proposals, we re-rank them with the full consistency score in Eq.2 and pick the top-K instances as the output knowledge.

We describe more nuanced handling in the search procedure (e.g., the processing of multi-token entities) in the appendix.

**Generalization to complex relations**    Most existing KGs or knowledge bases include relations that are predicates connecting two entities, e.g., `"A is capable of B"`. Yet, many relations in the real life can be more complex. Our approach is flexible and easily extensible to extract knowledge about those complex relations. We demonstrate in our experiments with two cases: (1) ***highly customized relations*** that have specific and sophisticated meaning, such as `"A is capable of, but not good at, B"`. Such sophisticated knowledge is often hard for human to manually write down massively. Our automatic approach naturally supports harvesting such knowledge given only an initial prompt and few seed entities; (2) ***n-ary relations*** involving more than 2 entities, such as `"A can do B at C"`. Our approach can straightforwardly be extended to deal with $n$-ary relations by generalizing the above compatibility score and search strategy accordingly to accommodate more than two entities.

**Symbolic interpretation of neural LMs**    The harvested knowledge tuples, as consistently recognized by the LM cross varying prompts, can be seen as the underlying "beliefs" (Stich, 1979; Hase et al., 2021) of the LM about the world. The interpretable symbolic tuples allow us to easily browse those beliefs, analyze the knowledge capability of the black-box LM, and make intrinsic comparisons between different LMs to understand the effect of diverse configurations, such as model sizes and pretraining strategies, as we showcased in the experiments.

## 4    EXPERIMENTS

We validate the effectiveness of the proposed knowledge harvesting framework with extensive experiments. We conduct human evaluation of our outcome KGs, and illustrate the effectiveness of the automatically created prompts in our framework. Besides, using our framework as a tool to interpret the LM knowledge storage, we make interesting observations about several knowledge-related questions on the black-box LMs.

We will release the whole family of extracted KGs, as well as the code implementation of our framework (included in the supplementary materials), upon acceptance.

### 4.1    SETUP

**Relations**    We extract knowledge about a large diverse set of relations. We collect relations from two existing knowledge repositories, and also test on a set of new relations not included in existing KGs. Specifically, our relations include (1) **ConceptNet** (Speer et al., 2017) is one of the most popular commonsense KGs and is widely used for evaluating knowledge extraction. Following Li et al. (2016), we filter the KG and subsample a set of 20 common relations (e.g. `has_subevent`, `motivated_by_goal`). The initial prompts for the relations are from the ConceptNet repository, and we randomly sample seed entity pairs from the ConceptNet KG for each relation. (2) **LAMA** (Petroni et al., 2019) is a popular benchmark for factual probing, containing knowledge tuples from multiple sources. Following the recent work (Jiang et al., 2020; Shin et al., 2020; Zhong et al., 2021), we use the T-REx split 41 Wikipedia relations, such as `capital_of`, `member_of`). For each relation, the human-written prompt provided in Petroni et al. (2019) is used as the initial prompt and we randomly sample seed entity pairs for each relation. (3) **New Relations**: To test the flexibility of our framework for harvesting novel knowledge, we write 14 new relations of interests that can hardly be found in any existing KGs, and manually write initial prompts and seed entity pairs for them. The resulting relations include complex relations as described in §3.2. More details, e.g., initial prompts, and seed entities of relations, can be found in the appendix.

| Head entity | Relation | Tail entity | Head entity | Relation | Tail entity |
|---|---|---|---|---|---|
| humidity | *prevent* | excessive temprature | viruses | *potential risk* | virus transmission |
| care | *prevent* | harm | prolonged sleep | *potential risk* | sleep disorders |
| local council | *can help* | village | serious offence | *potential risk* | conviction |
| therapist | *can help* | client | electricity | *ingredient for* | electric lamp |
| lake | *place for* | picnic tables | rice | *ingredient for* | soup |
| studios | *place for* | live shows | milk | *ingredient for* | butter |
| apple tree | *can but not good* | wood | locomotive | *can but not good* | speed trains |
| Relation: A can do B at C | (people, communicate, web) | | Relation: A needs B to C | (singers, vocal accompaniment, dance) | |
| Relation: A can do B at C | (adult couples, marry, marriage) | | Relation: A needs B to C | (human lives, survival, flourish) | |
| Relation: A can do B at C | (skier, ski downhill, mountain) | | Relation: A needs B to C | (actors, dialogue, portray characters) | |

Figure 2: Examples of knowledge tuples harvested from DISTILLBERT (Randomly sampled).

**Hyperparameters** We use 5 seed entity pairs for each relation. To automatically collect prompts (§3.1), we use GPT-3 with the instruction `"paraphrase:{sentence}"` as the off-the-shelf paraphraser. In the entity pair searching (§3.2), we restrict every entity to appear no more than 10 times to improve the diversity of generated knowledge and search out at most 50,000 entity tuples for each relation. We finally use various scoring thresholds to get the outcome KGs in different scales, including (1) **50%**: taking a half of all searched-out entities with higher consistency (Eq. 2). (2) **base-k**: Naturally there are different numbers of valid tuples for different relations (e.g. tuples of CAPITALOF should not exceed 200 as that is the number of all the countries in the world). We design a relation-specific thresholding method, that is to set 10% of the $k$-nd consistency as the threshold (i.e., $0.1 * \text{consistency}_k$), and retain all tuples with consistency above the threshold. We name the settings **base-10** and **base-100** when k is 10 and 100, respectively.

## 4.2 EVALUATION

### 4.2.1 EVALUATING OUTCOME KGS

We apply our framework to extract knowledge graph of **ConceptNet** (CN) relations and **New** relations from language models, and then conduct human evaluation on the accuracy of the extracted knowledge with Amazon MTurk. The correctness of each extracted knowledge tuple is labeled by 3 annotators with True/False/Unjudgeable. A tuple would be "accepted" (acc) if at least 2 annotators think it is true knowledge, and "rejected" (rej) if at least 2 annotators rate it as false. We provide more details of human evaluation in the appendix.

| Method | Tuple | Diversity | Novelty% | Acc% |
|---|---|---|---|---|
| WebChild | 4,649,471 | - | - | 82.0* |
| ASCENT | 8,600,000 | - | - | 79.2* |
| TransOMCS | 18,481,607 | 100,659 | 98.3 | 56.0* |
| COMET$_{base\text{-}10}^{CN}$ | 6,741 | 4,342 | 35.5 | 92.0 |
| COMET$_{50\%}^{CN}$ | 230,028 | 55,350 | 72.4 | 66.6 |
| ROBERTANET$_{base\text{-}10}^{CN}$ | 6,741 | 6,107 | 64.4 | 88.0 |
| ROBERTANET$_{base\text{-}100}^{CN}$ | 24,375 | 12,762 | 68.8 | 81.6 |
| ROBERTANET$_{50\%}^{CN}$ | 230,028 | 80,525 | 87.0 | 55.0 |
| ROBERTANET$_{base\text{-}10}^{New}$ | 2,180 | 3,137 | - | 81.8 |
| ROBERTANET$_{base\text{-}100}^{New}$ | 7,329 | 6,559 | - | 68.6 |
| ROBERTANET$_{50\%}^{New}$ | 23,666 | 16,089 | - | 58.6 |

Table 2: Statistics of KGs constructed with different methods. *Diversity* refers to the number of unique entities in a KG, and *Novelty* refers to the proportion of entities that don't appear in ConceptNet. The acceptance rate with * are from the original papers and subject to different evaluation protocol. Given COMET can only predict the tail entity given a source entity and a relation, we generate KGs with COMET by feeding it the head entity and relation of our RobertaNet. The first two blocks of this table corresponds to the first two blocks in Table 1, which includes three popular text mining methods: WebChild (Tandon et al., 2014), ASCENT (Nguyen et al., 2021), and TransOMCS (Zhang et al., 2020).

The statistics of the generated KGs from our framework (ROBERTANET) and other knowledge extraction methods are mentioned in Table 2, and the samples of the extracted KG from DISTILLBERT can be found in Figure 2. The acceptance rate, indicating the precision of KGs, in Table 2 are *not comparable* to each other, and only serves for a rough sense. From those statistics, we can see that with solely the LM as the knowledge source, and without any training data,

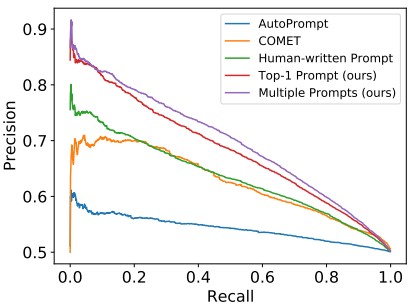

Figure 3: Precision-recall on ConceptNet relations.

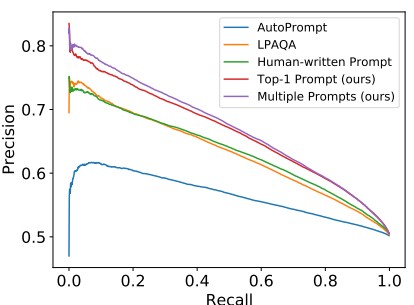

Figure 4: Precision-recall curve on LAMA relations.

| Methods | Acc | Rej |
|---|---|---|
| AUTOPROMPT | 0.33 | 0.47 |
| HUMAN PROMPT | 0.60 | 0.27 |
| TOP-1 PROMPT (Ours) | 0.69 | 0.23 |
| MULTI PROMPTS (Ours) | **0.73** | **0.20** |

Table 3: The portions of accepted and rejected tuples in human evaluation across settings, with the ROBERTA-LARGE as the LM.

| Source LMs | Acc | Rej |
|---|---|---|
| DISTILBERT | 0.67 | 0.24 |
| BERT-BASE | 0.63 | 0.26 |
| BERT-LARGE | 0.70 | 0.22 |
| ROBERTA-BASE | 0.70 | 0.22 |
| ROBERTA-LARGE | 0.73 | 0.20 |

Table 4: The portions of accepted and rejected tuples in human evaluation across different LMs, using the MULTI-PROMPTS approach.

our framework extracts KGs with competitive accuracy. Compared to COMET, our precision is a little lower, which is expected due to the big compromise we make in the setting, and our searching constraints ensure better diversity and novelty of ROBERTANET over COMET.

### 4.2.2 EVALUATING AUTOMATIC PROMPT CREATION

To evaluate the effect of our automatic creation of prompts, we compare the generated KGs under these settings of prompts on the human-written new relations: (1) **Multi-Prompts** refers to the the full framework described in §3 which use the automatically created diverse prompts in knowledge search. (2) **Top-1 Prompt**: To ablate the effect of ensembling multiple prompts, we evaluate the variant that uses only the prompt with largest weight (§3.1) for knowledge extraction. (3) **Human Prompt**: To further understand the effectiveness of the automatically created prompts, we assess the variant that uses the initial prompt (typically written by human) of each relation. (4) **AutoPrompt** (Shin et al., 2020), which was proposed to learn prompts by optimizing the likelihood of tail entity prediction on the training set. To fit in our setting, we adapt it to optimize the compatibility score (Eq.1) on the seed entity pairs. We omit other prompt tuning work (e.g., Zhong et al., 2021; Qin & Eisner, 2021) because they either are difficult to fit in our problem or require more training data and fail with only the few shot of seed entity pairs in our setting.

The human-annotated accuracy is shown in Table 3. Our TOP-1 PROMPT significantly improves the accuracy up to 9% over the HUMAN PROMPT, indicating our prompt searching algorithm can produce high quality prompts. MULTI-PROMPTS further improves the accuracy by around 4%, which means the combination of diverse prompts better capture the semantics of a relation. The method using the optimized prompt by AUTOPROMPT gives a lower accuracy than the one with the human or searched prompt, because the small set of seed knowledge tuples are insufficient for learning effective prompts for the desired relations.

Based on the results above, we move a step forward to see how the created prompts influence the subsequent module in the framework. Specifically, we evaluate if the automatically created prompts (§3.1) bring the consistency scoring (§3.2) better balance of knowledge accuracy (precision) and coverage (recall), by comparing various scoring methods on existing terms in ConceptNet and LAMA dataset. To be more detailed, we use the knowledge tuples from ConceptNet and LAMA as positive samples (§4.1), and synthesize the same amount of negative samples with the same strategy in Li et al. (2016) by random replacing entities or relations in the true knowledge. Each evaluated method estimates a score of each sample being positive (i.e., true knowledge), and ranks the samples based

on the scores. We can then compute both the *precision* and *recall* of positive samples at different cut-off points along the ranking, and plot the precision-recall curves for each method.

The automatic evaluation setting on given knowledge terms also enables us to adapt existing prevalent works, e.g., KG completion and factual probing (Table 1), for comparison with our approach: **(1) COMET** (Bosselut et al., 2019) is a transformer-based KG completion model trained to predict the tail entity $t$ conditioning on the head entity and relation $(h, r)$ on ATOMIC (Sap et al., 2019). The model has encoded rich commonsense knowledge. We thus test it on the ConceptNet commonsense data, and use its log-likelihood $\log P(t|h, r)$ as the score for each given knowledge tuple. **(2) LPAQA** (Jiang et al., 2020) collects a set of prompts on LAMA with text mining and paraphrasing, and ensembles them to optimize the factual probing task (and thus cannot extract new knowledge like ours). We report the best performance of its three variants.

The resulting precision-recall curves on ConceptNet and LAMA knowledge are shown in Figure 3 and Figure 4, respectively. The variant of our TOP-1 PROMPT from our automatically-collected prompt set is significantly better than the HUMAN PROMPT. This is because the human-written prompts often fail to effectively account for possible ambiguities in the expression, which are eliminated in our approach by leveraging the seed entity pairs with compatibility measure. Increasing the number of prompts (MULTI-PROMPTS) provides further improvements, showing the multiple automatically created prompts can represent the desired relations more accurately and better guide the LMs for consistency. Our approach also substantially outperforms other baselines, like COMET on ConceptNet and LPAQA on LAMA, indicating better knowledge correctness (precision) and coverage (recall) of our framework in the context. On both datasets, AUTOPROMPT, with only 5 seed entity pairs as its training data, gives inferior performance than other approaches, indicating that the current prompt optimization is not feasible for discovering knowledge of new relations without large training data.

### 4.3 ANALYSIS OF KNOWLEDGE IN DIFFERENT LMS

As discussed in §3, the outcome knowledge graphs can be treated as a fully symbolic interpretation of pretrained language models. We use MULTI-PROMPTS to harvest KGs from 5 different LMs and evaluate them with human annotation. The results are shown in Table 4, which sheds some new light on several knowledge-related questions regarding the LMs' knowledge capacity.

**Does a larger LM encode better knowledge?** For BERT (and RoBERTa), the large version and the base version share the same pretraining corpus and tasks, respectively, while the large version has a larger model architecture than the base version in terms of layers (24 v.s. 12), attention heads (16 v.s. 12), and the number of parameters (340M v.s. 110M). We can see that BertNet-large and RoBERTaNet-large are around 7% and 3% higher than their base version, separately, so the large models indeed encoded better knowledge than the base models.

**Does better pretraining bring better knowledge?** RoBERTa uses the same architecture as BERT but with better pretraining strategies, like dynamic masking, larger batch size, etc. In their extracted knowledge graphs from our framework, RoBERTaNet-large performs better than BertNet-large (0.73 v.s. 0.70), and RoBERTaNet-base is also better than BertNet-base (0.70 v.s. 0.63), which indicates the better pretraining indeed bring the better knowledge learning and storage.

**Is knowledge really kept in the knowledge distillation process?** DistilBERT is trained by distilling BERT-base, and reduces 40% parameters from it. Interestingly, the knowledge distillation process instead improves around 4% of accuracy in the result knowledge graph. This might be because the knowledge distillation is able to remove some noisy information from the teacher model.

## 5 CONCLUSION

We have developed an automatic framework that extracts a KG from a pretrained LM (e.g, BERT, ROBERTA), in an efficient and scalable way, resulting in a family of new KGs, which we refer to as BERTNET, ROBERTANET, etc. Our framework is capable of extracting knowledge of arbitrary new relation types and entities, without being restricted by pre-existing knowledge or corpora. The resulting KGs also serve as interpretation to the source LMs, bringing new insights of the knowledge capability in various LMs. Our current design and experimental studies are limited on LMs in the generic domain, and are not yet been studied in specific domains such as extracting healthcare knowledge from relevant neural models. We leave the exciting work of harvesting knowledge from various kinds of neural networks across applications and domains in the future work.

**Ethical considerations** In this work, the harvested knowledge is automatically generated by LMs. We would like to note that the language models could possibly generate unethical knowledge tuples, same with the risks of other applications using language models for generation. We hope that the knowledge extraction study could offer techniques to better interpret and understand the language models, and in turn foster the future research of language model ethics. Since the knowledge graph only consists simple phrases, we think filtering sensitive words would be effective. No foreseeable negative societal impacts are caused by the method itself.

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

## A APPENDIX

### A.1 COMPUTE RESOURCE

All of our experiments are running on a single Nvidia GTX1080Ti GPU. Harvesting a knowledge graph of one relation with Roberta-large takes about one hour.

### A.2 THE LICENSE OF THE ASSETS

All the data we used in this paper, including datasets, relation definitions, and seed entity pairs, etc., are officially public resources.

### A.3 PREPROCESSING OF CONCEPTNET

We filter out some linguistic relations (e.g. `etymologically derived from`) and some trivial relations (e.g. `related to`). We only consider the tuples with confidence higher than 1, and filter out relations comprising less than 1000 eligible tuples. We don't directly take the test set from Li et al. (2016) because they reserve a lot of tuples for training, resulting in a small and unbalanced test set.

### A.4 RELATION DEFINITIONS

The initial prompts of ConceptNet are from its repository [1]. For LAMA relations, we use the human written "template" for every relation as the initial prompt, which are all included in the LAMA dataset. All the **New relations**, together with the generated prompts, are shown in Table 5 and the list below.

### A.5 PROCESSING OF MULTI-TOKEN ENTITIES

Each entity may take more than one BPE tokens when conducting the mask-filling task. In our entity searching step, before searching the tokens to fill the mask, we enumerate the number of masks for every entity and in our setting, each entity has one or two BPE tokens. We showcase the likelihood calculation of multiple tokens in Figure 7.

In the candidate entity pairs proposal step, we use the minimum individual log-likelihoods (shorted as MLL) instead of the full Equation 2, which allows us to apply a pruning strategies. For example, when we are searching for 100 entity tuples, we maintain a minimum heap to keep track of the MLL of the existing entity pair set. The maximum size of this heap is 100, and the heap top can be used as a threshold for future search because it's the $100\text{-}th$ largest MLL: When we are searching for a new entity tuple, once we find the log likelihood at any time step is lower than the threshold, we can prune the continuous searching immediately, because this means the MLL of the this tuple will never surpass any existing tuples in the heap. If a new entity tuple is searched out without being pruned, we will pop the heap and push the MLL of the new tuple. Intuitively, the pruning process makes sure that the generated part of the tuple in searching is reasonable for the given prompt.

### A.6 HUMAN EVALUATION

The wage is $0.15 for each three questions. The average time for answering a question is about 15 seconds, so the estimated wage of an hour is $12 per hour. The total cost on human annotation

---

[1]https://github.com/commonsense/conceptnet5/wiki/Relations (the "Description" column.)

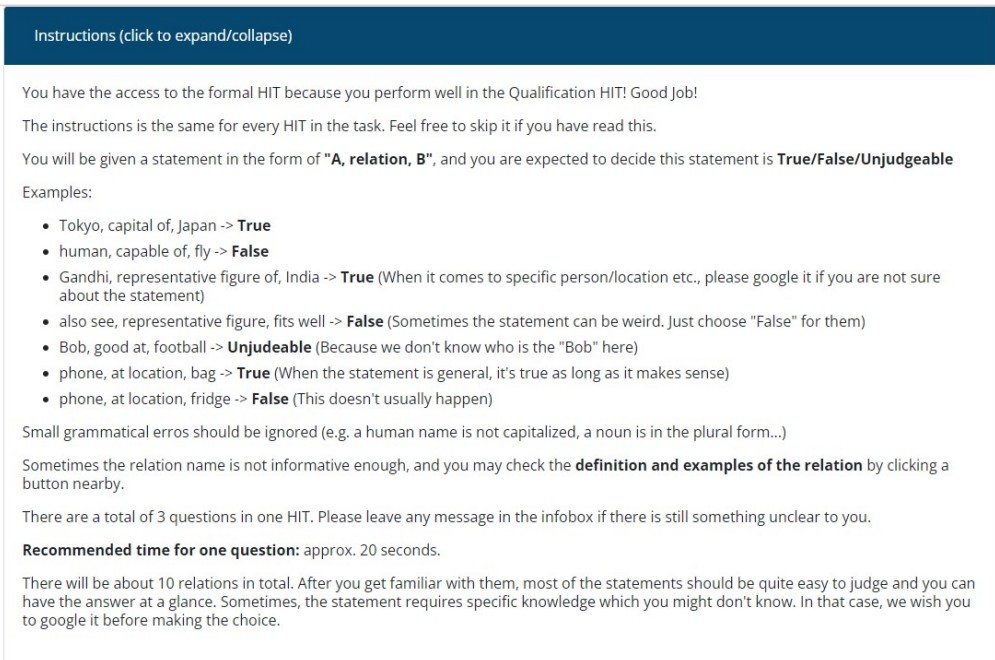

Figure 5: The instruction to annotators

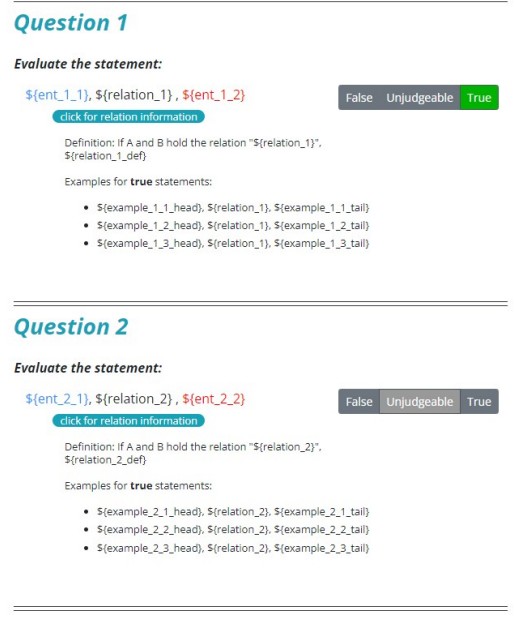

Figure 6: The questions to annotators

is $1580. We present the screenshot of the instruction in Figure 5 and question in Figure 6. The inter-annotator agreement (Krippendorff's Alpha) is 0.27, showing fair agreement.

| Relation | Initial prompts | Seed entity pairs |
| --- | --- | --- |

| somebody do something at | $ENT_0$ can $ENT_1$ in $ENT_2$ | (people, work out, gym)
(bird, fly, sky)
(student, study, classroom)
(player, play, ground)
(sodier, fight, battleground) |
|---|---|---|
| prevent | $ENT_0$ prevents $ENT_1$ | (mask, virus)
(exercise, disease)
(study hard, fail the exam)
(reading, stupid)
(insurance, bankruptcy) |
| help | $ENT_0$ can help $ENT_1$ | (doctor, patient)
(teacher, student)
(housekeeper, housewife)
(teaching assistant, professor)
(police, victim) |
| place for | $ENT_0$ is the place for $ENT_1$ | (gym, exercise)
(classroom, study)
(office, work)
(hosipital, medical treatment)
(mart, shopping) |
| antonym | $ENT_0$ is the opposite of $ENT_1$ | (fat, thin)
(happy, sad)
(north, south)
(woman, man)
(pass, fail) |
| separated by the ocean | $ENT_1$ and $ENT_0$ are separated by the ocean | (China, Japan)
(Australia, New Zealand)
(United Kingdom, the Continent)
(United State, Cuba)
(Spain, Morocco) |
| ingredient for | $ENT_0$ is an ingredient for $ENT_1$ | (flour, cake)
(beef, hamburger)
(potato, chip)
(fried rice, rice)
(stargazer pie, pilchard) |
| source of | $ENT_0$ is the source of $ENT_1$ | (wool, woollen sweater)
(milk, yogurt)
(sand, silicon)
(iron mine, iron)
(crude oil, fuel) |
| business | $ENT_0$ sells $ENT_1$ | (Nissan, car)
(Apple, laptop)
(Shell, oil)
(Nvidia, GPU)
(McDonald's, hamburger) |
| featured thing | $ENT_0$ is a very $ENT_1$ $ENT_2$ | (egg, cheap, food)
(boa, long, snake)
(Messi, skillful, football player)
(Russia, large, country)
(Amazon, rich, company) |
| need sth to do sth | $ENT_0$ needs $ENT_1$ to $ENT_2$ | (developer, computer, code)
(people, social media, connect)
(pig, food, grow)
(people, money, live)
(intern, good performance, return offer) |
| can but not good | $ENT_0$ can $ENT_1$ but not good at | (chicken, fly)
(dog, swim)
(long-distance runner, sprint)
(skater, ski)
(researcher, teach) |

| worth celebrating | It's worth celebrating for a $ENT_0$ to $ENT_1$ | (team, wim)
(student, pass exam)
(researcher, publish paper)
(people, earn money)
(parents, get a baby) |
|---|---|---|
| potential risk | A potential risk of $ENT_0$ is $ENT_1$ | (playing game, fail the exam)
(sleep insufficiency, heart disease)
(candy, tooth decay)
(flight, plane crash)
(investment, lose money) |

Table 5: prompt and entity pairs of new relations

The generated prompts are listed below:

1. **somebody do something at**:
   at the $ENT_2$, $ENT_0$ can $ENT_1$
   $ENT_0$s can $ENT_1$ in the $ENT_2$
   the $ENT_2$ is a place where $ENT_0$ can $ENT_1$
   the $ENT_0$ can choose to $ENT_1$ in the $ENT_2$ or online
   a $ENT_0$ can learn effectively by $ENT_1$ing in a $ENT_2$ setting
   looking up at the $ENT_2$, it's easy to see that $ENT_0$s can $ENT_1$
   at the $ENT_2$, $ENT_0$s can $ENT_1$ together and learn from each other
   a $ENT_0$ can $ENT_1$ in a $ENT_2$ by themselves or with a group of friends
   at the $ENT_2$, $ENT_0$ can use the machines to $ENT_1$, or they can take classes
   at the $ENT_2$, $ENT_0$ can $ENT_1$ by using the various machines and equipment available
   there are many facilities available for $ENT_0$ who want to stay in shape and $ENT_1$, one of which is the $ENT_2$

2. **prevent**:
   $ENT_0$ prevents $ENT_1$
   if you $ENT_0$, you will not $ENT_1$
   $ENT_0$ provides a safety net against $ENT_1$
   if you $ENT_0$, you are less likely to $ENT_1$
   a $ENT_0$ will protect you from getting the $ENT_1$
   if you want to avoid being $ENT_1$, start $ENT_0$ more
   if you don't want to be $ENT_1$, you should $ENT_0$ often
   $ENT_0$s are effective at preventing the spread of $ENT_1$es
   this is a paraphrase of the saying $ENT_0$ prevents $ENT_1$ity.
   $ENT_0$ is an important part of staying healthy and preventing $ENT_1$
   regular $ENT_0$ has been shown to be one of the most effective ways to prevent $ENT_1$
   $ENT_0$ protects people from having to declare $ENT_1$ in the event of an accident or emergency
   $ENT_0$s prevent the spread of $ENT_1$es by trapping droplets that are released when the user talks, coughs, or sneezes
   it could be said that $ENT_0$ prevents $ENT_1$, as it provides a safety net for people in the event of an unexpected setback
   regular $ENT_0$ has various health benefits and is often prescribed by doctors as a preventative measure against developing various $ENT_1$s

3. **help**:
   a $ENT_0$ can help a $ENT_1$
   the $ENT_0$ can help with the $ENT_1$'s chores
   a $ENT_0$ can assist a $ENT_1$ with her duties
   if you are a $ENT_1$ of a crime
   the $ENT_0$ can help you
   the $ENT_0$ can help the $ENT_1$ with their medical needs
   a $ENT_0$ can help a $ENT_1$ by providing guidance and support
   a $ENT_0$ can help take care of the household duties for a $ENT_1$
   a $ENT_0$ officer can help a $ENT_1$ by providing them with protection and assistance

a $ENT_0$ can help with the chores around the house, giving the $ENT_1$ more free time

a $ENT_0$ can help out a $ENT_1$ by doing things like cleaning, cooking, and running errands

a $ENT_0$ can help a $ENT_1$ in many ways, from grading papers to leading discussion sections

a $ENT_0$ can help a $ENT_1$ by providing guidance, answering questions, and offering feedback

a $ENT_0$ can help a $ENT_1$ with their workload by taking on some of the teaching responsibilities

a $ENT_0$ can help a $ENT_1$ by grading papers, leading discussion sections, and providing office hours

the $ENT_0$ can help $ENT_1$s of crime by investigating the incident and taking statements from witnesses

4. **place for**:

the $ENT_0$ is the place for $ENT_1$

there's no place like $ENT_0$ for $ENT_1$

$ENT_0$ is a great place to do some $ENT_1$

the $ENT_0$ is a place where people go to $ENT_1$

there's no better place to $ENT_1$ than in an $ENT_0$

the $ENT_0$ is the best place to get some $ENT_1$ done

the $ENT_0$ provides an environment for people to $ENT_1$

if you want to do some $ENT_1$ing, you can go to the $ENT_0$

if you're looking to do some $ENT_1$, $ENT_0$ is the place for you

if you're looking to do some $ENT_1$, then you should head on over to $ENT_0$

if you're looking to get fit and $ENT_1$, the $ENT_0$ is the perfect place to do so

there's no place like the $ENT_0$ for $ENT_1$ because you have access to all the equipment you need to get a good workout

5. **antonym**:

a $ENT_0$ is not a $ENT_1$

the antonym of $ENT_0$ is $ENT_1$

$ENT_1$ is the opposite of $ENT_0$

when you're $ENT_1$, you're not $ENT_0$

$ENT_1$ is to $ENT_0$ as night is to day

$ENT_0$ and $ENT_1$ are opposite concepts

if you $ENT_1$, it means you didn't $ENT_0$

when a person is $ENT_0$, they are not $ENT_1$

the two words "$ENT_0$" and "$ENT_1$" are antonyms

$ENT_0$ is the direction that is opposite of $ENT_1$

to be $ENT_1$ is to have a small amount of body $ENT_0$

when something $ENT_0$es, it is successful, and when something $ENT_1$s, it is not successful

if you $ENT_0$ something, you have succeeded, and if you $ENT_1$ something, you have not succeeded

this means that simply because someone does not $ENT_0$ something does not mean they have $ENT_1$ed

6. **separated by the ocean**:

the ocean separates $ENT_1$ from the $ENT_0$

there is an ocean separating $ENT_1$ and $ENT_0$

the pacific ocean lies between $ENT_0$ and $ENT_1$

$ENT_1$ and $ENT_0$ are separated by the tasman sea

there is a distance of about 1,500 miles between $ENT_1$ and the $ENT_0$s

the distance between $ENT_1$ and $ENT_0$ is vast, with an ocean between them

$ENT_1$ and $ENT_0$ are two different countries located on different continents

$ENT_0$ and $ENT_1$ are two countries that are close to each other but are separated by the ocean

there is a large body of water, known as the atlantic ocean, which separates the two land masses of $ENT_1$ and the $ENT_0$

while $ENT_1$ and $ENT_0$ are both located in the oceania region of the world, the two countries are separated by the tasman sea

7. **ingredient for**:

$ENT_1$ needs $ENT_0$ as an ingredient

one way to use $ENT_0$es is to make $ENT_1$s
$ENT_0$ is a key ingredient in making a $ENT_1$
one of the ingredients for making a $ENT_1$ is $ENT_0$
if you want to make a $ENT_1$, you'll need some $ENT_0$
a $ENT_0$ is an ingredient that is used to make $ENT_1$s
$ENT_0$ is a dish that contains $ENT_1$ as one of its ingredients
$ENT_0$ is a popular dish made with $ENT_1$ as the main ingredient
$ENT_0$ is a type of fish pie that is typically made with $ENT_1$s
$ENT_0$ is a dish made from $ENT_1$ that has been fried in a wok or a pan

8. **source of**:
$ENT_1$ comes from $ENT_0$
$ENT_0$ is used to make $ENT_1$
most $ENT_1$ is made from $ENT_0$
$ENT_0$ is a major source of $ENT_1$
the main source of $ENT_1$ is $ENT_0$
the $ENT_0$ is what turns into $ENT_1$
$ENT_1$ is most commonly found in $ENT_0$s
the $ENT_0$ is the place where $ENT_1$ is mined
many $ENT_1$s contain $ENT_0$ as a major ingredient
$ENT_0$s are where $ENT_1$ is pulled from the ground
$ENT_1$ is a dairy product that is made from $ENT_0$
$ENT_0$ is the primary ingredient in the production of $ENT_1$
the main source of $ENT_1$ is quartz, which is a type of $ENT_0$
$ENT_0$ is the main source of $ENT_1$ used in vehicles and other machinery
$ENT_0$ is used to produce gasoline, diesel, and other petroleum-based products, which are widely used as $ENT_1$ for cars, trucks,

9. **business**:
$ENT_0$ sells $ENT_1$s
$ENT_0$ is an $ENT_1$ company
$ENT_0$ is known for selling $ENT_1$s
$ENT_0$ is a company that sells $ENT_1$s
$ENT_0$ is an international $ENT_1$ and gas company
$ENT_0$ is a $ENT_1$ company that manufactures and sells vehicles
$ENT_0$ sells $ENT_1$s through its online store and retail locations
$ENT_0$ is a company that sells graphics processing units ($ENT_1$s)
$ENT_0$ is a popular fast food chain that is known for selling $ENT_1$s
$ENT_0$ is ajapanese$ENT_1$ manufacturing company that is headquartered in yokohama, japan
$ENT_0$ is a fast food company that specializes in $ENT_1$s, fried chicken, and soft drinks
$ENT_0$ is a company that specializes in the production of $ENT_1$s, or graphics processing units
$ENT_0$ is a japanese $ENT_1$ company that sells a wide variety of vehicles, from small to large, and from economy to luxury
$ENT_0$ describes themselves as "the worldu2019s largest information technology company by revenue," and they sell many products, including $ENT_1$s

10. **featured thing**:
the $ENT_0$ $ENT_2$ is very $ENT_1$
$ENT_0$s are a very $ENT_1$ $ENT_2$
$ENT_0$ is definitely a $ENT_1$ $ENT_2$
$ENT_0$ is a very $ENT_1$ and successful $ENT_2$
you can hardly find a $ENT_1$er $ENT_2$ than $ENT_0$s
there is no doubt that $ENT_0$ is a very $ENT_1$ $ENT_2$
it covers a lot of ground, $ENT_0$ is a very $ENT_1$ $ENT_2$
a single $ENT_0$ is a very $ENT_1$ and affordable $ENT_2$ item
$ENT_0$s are a type of $ENT_2$ that can grow to be very $ENT_1$
although $ENT_0$ is not the world's $ENT_1$est $ENT_2$, it is still very wealthy

11. **need sth to do sth**:
$ENT_0$ need $ENT_1$ to $ENT_2$

in order to $ENT_2$, $ENT_0$ need $ENT_1$
there is a need for $ENT_1$ to $ENT_2$ $ENT_0$
a $ENT_0$ requires $ENT_1$ in order to $ENT_2$
$ENT_0$ feel the need to $ENT_2$ through $ENT_1$
a $ENT_0$ requires $ENT_1$ to $ENT_2$ with each other
$ENT_0$ feel that they need $ENT_1$ in order to $ENT_2$
a $ENT_0$ will only $ENT_2$ if it has enough $ENT_1$ to eat
without $ENT_1$, $ENT_0$ would have a harder time $ENT_2$ing with others
it is important for $ENT_0$ to be able to $ENT_2$ with each other through $ENT_1$

12. **can but not good**:
$ENT_0$s are not good at $ENT_1$ing
$ENT_0$s can $ENT_1$, but they aren't very good at it
the average $ENT_0$ is not especially good at $ENT_1$ing
the $ENT_0$ is skilled at $ENT_1$ing, but not particularly good at it
$ENT_0$s are not good at $ENT_1$ing because they are not built for it
$ENT_1$ing requires explosive speed and power, which $ENT_0$s typically lack
$ENT_0$s aren't particularly good $ENT_1$mers, but they can do it if they need to
while $ENT_0$s are capable of $ENT_1$ming, they are not particularly proficient at it
$ENT_0$s are not good at $ENT_1$ing because they lack the proper equipment and training
while $ENT_0$s are typically excellent at their jobs, $ENT_1$ing is not usually a strong suit
although $ENT_0$s are theoretically able to $ENT_1$, they are not very good at it and usually stay on the ground
while $ENT_0$s are able to $ENT_1$, they are not instinctively good at it and may need some help or encouragement to do so

13. **worth celebrating**:
it's good when $ENT_0$ $ENT_1$
it's great for $ENT_0$ to $ENT_1$
it's great when a group of $ENT_0$ $ENT_1$
it's great when $ENT_0$s $ENT_1$s as a group
it's worth celebrating when a $ENT_0$ $ENT_1$
it's always good when $ENT_0$ $ENT_1$ is doing well
when a group of $ENT_0$ $ENT_1$, it is worth celebrating
there is value in celebrating when group of $ENT_0$s $ENT_1$
it's always good when $ENT_0$s $ENT_1$s, regardless of the topic
it is seen as significant accomplishment for $ENT_0$ when they $ENT_1$

14. **potential risk**:
$ENT_0$ may lead to $ENT_1$
$ENT_0$ can potentially cause $ENT_1$
one potential risk of $ENT_0$ is $ENT_1$
eating too much $ENT_0$ can cause $ENT_1$
if a $ENT_1$es, it is a potential risk of $ENT_0$
if you have $ENT_1$, you may be at risk for $ENT_0$
$ENT_0$ has been linked to an increased risk of $ENT_1$
an $ENT_0$ always entails some riskŽ2014you could $ENT_1$ on it
$ENT_0$ can lead to $ENT_1$ if it is not eaten in moderation
an $ENT_0$ always comes with the potential to make or $ENT_1$
if a $ENT_0$ is not properly executed, it can cause a $ENT_1$
if a $ENT_0$ is not well-planned or executed, it may lead to a $ENT_1$
a new study has found that $ENT_0$ is linked to an increased risk of $ENT_1$
if you have a $ENT_1$, you are at a greater risk for cavities if you eat $ENT_0$
there is a strong correlation between $ENT_0$ and an increased risk of air$ENT_1$es
a recent study has found that there is a correlation between eating $ENT_0$ and an increased risk of $ENT_1$

## A.7 Detailed results of human evaluation

We show the detailed results of human evaluation in Table 6.

Table 6: Detailed result of human evaluation. The numbers indicate the portions of accepted and rejected tuples. Ro-l, DB, B-b, B-l, Ro-b are short for Roberta-large, DistilBert, Bert-large, Bert-base, Roberta-base. Human, Auto, Top-1 and Multi stand for methods that use Human Prompt, Autoprompt, Top-1 Prompt (Ours) and Multi Prompts (Ours).

| Model | Ro-l | Ro-l | Ro-l | Ro-l | DB | B-b | B-l | Ro-b |
|---|---|---|---|---|---|---|---|---|
| Prompt | Human | Auto | Top-1 | Multi | Multi | Multi | Multi | Multi |
| BUSINESS | 0.60/0.32 | 0.76/0.13 | 0.75/0.16 | 0.88/0.07 | 0.54/0.27 | 0.64/0.23 | 0.76/0.13 | 0.74/0.19 |
| HELP | 0.77/0.12 | 0.52/0.34 | 0.92/0.03 | 0.87/0.05 | 0.91/0.04 | 0.81/0.04 | 0.88/0.06 | 0.88/0.06 |
| INGREDIENT FOR | 0.59/0.33 | 0.33/0.59 | 0.73/0.20 | 0.71/0.24 | 0.70/0.26 | 0.55/0.40 | 0.72/0.23 | 0.51/0.40 |
| PLACE FOR | 0.76/0.10 | 0.41/0.36 | 0.63/0.32 | 0.89/0.07 | 0.84/0.14 | 0.78/0.18 | 0.87/0.11 | 0.88/0.09 |
| PREVENT | 0.42/0.42 | 0.18/0.67 | 0.60/0.25 | 0.40/0.45 | 0.60/0.32 | 0.44/0.39 | 0.62/0.25 | 0.68/0.25 |
| SOURCE OF | 0.76/0.17 | 0.21/0.67 | 0.52/0.44 | 0.60/0.33 | 0.63/0.36 | 0.65/0.32 | 0.75/0.24 | 0.55/0.37 |
| SEPARATED BY THE OCEAN | 0.48/0.38 | 0.16/0.48 | 0.56/0.35 | 0.55/0.40 | 0.51/0.24 | 0.57/0.26 | 0.44/0.46 | 0.44/0.49 |
| ANTONYM | 0.50/0.41 | 0.10/0.83 | 0.50/0.48 | 0.55/0.44 | 0.38/0.56 | 0.41/0.56 | 0.52/0.42 | 0.75/0.22 |
| FEATURED THING | 0.85/0.12 | 0.38/0.40 | 0.88/0.06 | 0.89/0.10 | 0.37/0.44 | 0.44/0.40 | 0.46/0.44 | 0.65/0.20 |
| NEED A TO DO B | 0.71/0.18 | 0.62/0.21 | 0.66/0.22 | 0.79/0.10 | 0.83/0.12 | 0.62/0.25 | 0.65/0.18 | 0.72/0.17 |
| CAN BUT NOT GOOD AT | 0.52/0.34 | 0.29/0.42 | 0.61/0.19 | 0.44/0.21 | 0.51/0.31 | 0.60/0.21 | 0.64/0.22 | 0.39/0.35 |
| WORTH CELEBRATING | 0.47/0.29 | 0.23/0.51 | 0.81/0.05 | 0.85/0.08 | 0.79/0.12 | 0.74/0.14 | 0.84/0.10 | 0.83/0.10 |
| POTENTIAL RISK | 0.40/0.23 | 0.31/0.45 | 0.70/0.21 | 0.76/0.19 | 0.87/0.05 | 0.66/0.22 | 0.72/0.16 | 0.79/0.08 |
| A DO B AT | 0.56/0.33 | 0.14/0.55 | 0.79/0.14 | 0.97/0.03 | 0.93/0.07 | 0.93/0.05 | 0.94/0.06 | 0.94/0.06 |
| AVERAGE | 0.60/0.27 | 0.33/0.47 | 0.69/0.22 | 0.73/0.20 | 0.67/0.24 | 0.63/0.26 | 0.70/0.22 | 0.70/0.22 |

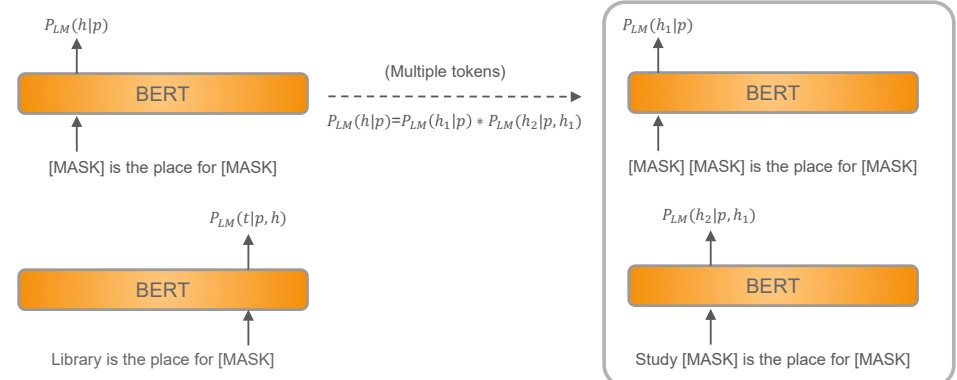

Figure 7: We demonstrate the calculation with an example where $p =$"A IS THE PLACE FOR B". The left two figures shows how we calculate $P_{LM}(h|p)$ and $P_{LM}(t|p,h)$. In this example, $h =$"library" when we set both head and tail entities to have one single token. The right block shows how we calculate the conditional probability of multiple-token entities by decomposing it into two steps. In this example, the first token of the head entity $h_1 =$"study".

