# OpenReview forum: "BertNet: Harvesting Knowledge Graphs from Pretrained Language Models"
_ICLR.cc/2023/Conference — Submitted to ICLR 2023_

### Official Review · Reviewer_dt4V · 2022-10-24

**Confidence:** 4
**Clarity, Quality, Novelty And Reproducibility:** N/A
**Correctness:** 3
**Technical Novelty And Significance:** 3
**Empirical Novelty And Significance:** 3
**Recommendation:** 5

**Details Of Ethics Concerns:**

There is a risk related to the bias that can be propagated from the pre-trained LM. The filter was not discussed in this work.

**Strength And Weaknesses:**

Strength:
The proposed idea is simple and seems straightforward to implement. The proposed method needs minimal human input (only an initial prompt and 5 candidate entity pairs for each relation). It effectively leverages GPT3 to rephrase and generate better alternative prompts used for extracting new entity pairs. Overall, the paper is well written with detailed examples to illustrate the main idea.

Weaknesses:
Although the model shows better results than the baseline (COMET), it still lacks more analysis in the evaluation to convince the effectiveness of the proposed model.

First, the result in Table 2 shows the precision of the prediction but not the recall. For example, the recall can be estimated on the known set of factual relations. The P-R curves in Figures 3 and 4 seem to partly answer this question. However, looking at the 0.8 level of precision, the recall is only less than 0.2, which raises another question about the overall effectiveness of the extraction method.

The baseline WebChild in Table 2 seems to extract a much higher number of tuples at 4.6M, while the proposed model ROBERTANET only extract 6741 tuples. It can be WebChild having much higher coverage, but this difference was not discussed in the paper.

Second, the quality of the extraction heavily depends on the compatibility scoring function (Eq. 1). It would be better to evaluate more on the design choice of this formula and explain why the proposed formula is optional for this purpose.

Third, Looking at the extracted relations, it looks like some relations are more non-trivial than others. It would be useful to analyze more on the factual relations (e.g., the relations between named entities such as 'ceo_of', 'invent') because these relations will be more useful for downstream applications like QnA. The LAMA dataset has more of these relations so authors can extract and analyze more case studies from this dataset.

Another suggestion, in the GPT3 prompt: "paraphrase: {sentence}" used to extend the prompt set, would it be better to include example of all 5 seed entity pairs into the prompt to help eliminating the ambiguity?


**Summary Of The Paper:**

Authors proposed a method to extract knowledge graph relations from a pretrained LM through automatic prompt creation and leveraging the pretrained LM to score the candidate entity pairs. Compared to previous works that usually rely on human annotated data or existing massive KGs, the authors' approach requires only the minimal definition of relations as inputs.

**Summary Of The Review:**

Overall, the idea proposed in this work is simple, easy to implement and it shows promising results compared to the baselines. However, the evaluation currently lacks analysis about the coverage and quality of the extracted relations.

---

> ### Author Response · Authors · 2022-11-10
> **Response**
>
> Thank you for your appreciation that our paper is well written, our idea is clearly illustrated and straightforward to implement, and we show promising results.
>
> ### Evaluation
>
> Regarding the questions on our evaluation, we would like to emphasize that, since we propose a novel task that has not been explored by previous papers, the evaluation is not a trivial job. **Human evaluation is the only direct evaluation in our work.**
>
> About the results in Table 2, as we mentioned in Sec. 4.2.1, WebChild, and all the other methods in Table 2 are **not directly comparable** to our framework, because, as we summarize in Table 1, they belong to other styles of knowledge extraction with different knowledge sources under different settings. Specifically, WebChild makes use of large-scale corpus on the Internet and even an existing KG – WordNet, which leads to an inner drawback that the framework only works on a small predefined set of relations. Correspondingly, our framework which doesn’t rely on any resources other than the LM, thus can be applied to any relations of interest. In Table 2, we apply our framework to widely-used ConceptNet relations ($\text{RobertaNet}^{CN}$ showing its effectiveness), and New Relations ($\text{RobertaNet}^{New}$ showing its generalizability). The scale of our framework presented in Table 2 doesn’t indicate the limit of our framework.
>
> About the effectiveness of the extraction method shown in the P-R curves. We would like to clarify that the comparison among methods is our focus, instead of the absolute values, because it’s highly subject to the dataset (e.g. the quality/attribute of reference KGs, the strategy of negative sampling, etc.). For example, in Figure 3 on ConceptNet relations, our method, without any training, outperforms COMET which has been trained on ConceptNet.
>
> In terms of the recall on existing KGs, while there are some previous works taking it as an evaluation metric of coverage [1, 2], there are also works considering high coverage of known KGs as a flag of inability to generate novel knowledge [3, 4], and the main goal of this framework is to extract knowledge of arbitrary relations, which doesn’t exist in any previous KGs, thus we tend to think the relative recall is not an appropriate metric to evaluate our framework.
>
> ### Scoring Function
>
> We briefly discuss the insights in Sec. 3.1 before presenting Equation 1. To be more specific, a natural way we can query masked LMs is to autoregressively fill the entity pair into the blanks, which is the first part of Equation 1. Besides the plain idea to add the log likelihood of each slot, we also come up to calculate the minimum of the likelihood, with the intuition that any valid entity pair should have neither too low p(h|p) nor low p(t|h, p), which indicates wrong entity type or wrong relation respectively.
>
> To specify the hyperparameter $\alpha$ in Equation 1, since it’s expensive to evaluate the KGs generated with different weights, we briefly tuned the weights under the setting of Figure 3 & 4, where we construct datasets of known entity pairs and hypothesize the performance on this task reflects the effectiveness of the scoring function.
>
> It is an interesting problem to explore more classes of the compatibility score in future work, but in this paper we focus on the principled framework and consider this problem as a design choice in the implementation. Thanks for raising this question.
>
> ### Choice of relations
> In this paper we focus more on commonsense knowledge, which usually attracts more attention from the research community and is also useful in many downstream NLP tasks [5, 6, 7]. There are some simple factual relations like “Business: Apple, iPhone” included in our “New relations”. Besides, we tested the scoring module on LAMA in Figure 4, indicating our framework still outperforms previous prompt creation methods on factual relations.
>
> We agree that extracting knowledge of factual relations is exciting and useful, but it would pose more challenges to conducting our large-scale human evaluation, which requires more knowledgeable human evaluators. We leave this challenging work for future research.
>
>
> [1] Advanced Semantics for Commonsense Knowledge Extraction, WWW 2021 \
> [2] Commonsense Properties from Query Logs and Question Answering Forums, CIKM 2019 \
> [3] COMET: Commonsense Transformers for Automatic Knowledge Graph Construction, ACL 2019 \
> [4] TransOMCS: From Linguistic Graphs to Commonsense Knowledge, IJCAI 2020 \
> [5] Enhancing Zero-shot and Few-shot Stance Detection with Commonsense Knowledge Graph , Findings of ACL 2021 \
> [6] Generalized Zero-shot Intent Detection via Commonsense Knowledge, SIGIR 2021 \
> [7] Knowledge Enhanced Reflection Generation for Counseling Dialogues, ACL 2022

---

> > ### Author Response · Authors · 2022-11-10
> > **Response (cont'd)**
> >
> > ### About GPT-3
> > Thanks for your suggestion. We did try out other prompts to GPT-3 in our preliminary experiments, and it turned out that paraphrasing is not a bottleneck of our framework, i.e. the current use of GPT-3 is enough to generate the prompts we need for the next stages.
> >
> > ### Ethics Concerns
> > We stated in the appendix that a word filter will be used to reduce ethical concerns. This is the prevalent method to filter unethical contents in texts on multiple online platforms. In our preliminary test, we take a commonly-used resource (https://github.com/LDNOOBW/List-of-Dirty-Naughty-Obscene-and-Otherwise-Bad-Words) and find it can effectively filter out some inappropriate contents in the generated KG.
> > We are aware and concerned about the ethical problems of language models, and we want to note that applying word filters is not a fundamental method to eliminate ethical risks. However, since ethical concerns is not the main topic we present in this paper, we take a relatively simple yet effective method and leave the important work for future research .

---

### Official Review · Reviewer_pDgC · 2022-10-25

**Confidence:** 4
**Correctness:** 3
**Technical Novelty And Significance:** 2
**Empirical Novelty And Significance:** 2
**Recommendation:** 3

**Clarity, Quality, Novelty And Reproducibility:**

Clarity: Some important technical details are missing, e.g., efficiently generating candidate entities. Please refer to the Summary of the review Q3.

Quality: The model is only evaluated by crowdsourcing, with no objective evaluation. The effectiveness is not significant compared with SOTA baselines.

Novelty: The idea is generally interesting, the authors bring the old wisdom of iterative KG construction into the pre-trained LM fashion. But the core technical contribution is just an LM-based reweighting. Methodological nvoelty is a shortage.

Reproducibility: Unable to evaluate, since the technical details are unclear.


**Strength And Weaknesses:**

Strength:
1. The problem of automatically constructing KG from pre-trained LMs is well-motivated.
2. It is interesting to combine the iterative KG construction into the current pre-trained LMs fashion.

Weakness:
1. The framework of iterative KG construction (i.e., bootstrapping) is not novel.
2. Some important technical details, e.g., how to generate multi-token entities and how to reduce the candidate set, are missing.
3. The experiment is not comprehensive. There is no objective evaluation.


**Summary Of The Paper:**

The paper proposes a pre-trained LMs based KG construction model aiming at identifying new entities for a single query relation. The proposed model is a bootstrapping strategy that starts from a query prompt with a few seed entity pairs and iteratively generates new pairs by means of new prompts generated by a GPT-based paraphrase model. Further, an LM-based reweight function is employed to weigh the new prompts and new pairs. The experiments conducted on crowdsourcing-based evaluation and case study to show the model is competitive to SOTA baselines.

**Summary Of The Review:**

1. The proposed framework is a kind of bootstrapping-based knowledge graph construction approach, which, reminisce of the Hearst patterns for hypernym relation. My major concern is how to effectively handle the semantic drift, i.e., the semantics of new patterns and instance harvest may change drastically along with iteration or paraphrasing. But authors did not discuss this problem in-depth or provide a method with a plausible bound. What we have in the model is the LM-based paraphrasing and weighting, however, the key point is, LM does not necessarily guarantee the semantics.

2. For the diversity of paraphrase (second paragraph of Section 3.1), it is not clear (1) why to use a textual measurement (i.e., edit distance) to ensure diversity of paraphrases (second paragraph of Section 3.1). It is a bit weird that the whole paper is deep learning based, except for the paraphrase generation, which adopts an old-fashioned textual similarity ED, which, is obviously out of the deep learning fashion. (2) how this ED constraint works. As ED works for a text pair, do all generated prompt pairs need to keep a certain distance?

3. Some important technical details are missing:
  (1) Efficient generation candidate entity pair generation of Section 3.2. Appendix A.5 has a few sentences talking about "mask-filling" but I cannot find the details of how to generate multi-token entities and how to use "thresholding and pruning" to reduce the candidate set.
  (2) For computing Eq.1, it is not clear (a) how to generate the query text. Is the query text formed by inserting a specific entity (e.g., insert h into p for P(h|p))? If so, what is the difference between Plm(t|p,h) and Plm(h,t|p)? (b) how to compute the probability of Plm(h,t|p), Plm(h|p), and Plm(t|p,h). Is the probability computed by the generative probability of the LM?


4. In addition to the manual evaluation (crowdsourcing rating), the objective evaluation based on evaluation datasets should be given. For example, the authors can select datasets adopted by relevant tasks, e.g., knowledge graph completion, such as WRNN, FB15K-237, etc. for evaluation.

5. The baseline methods should be introduced in a few sentences.

6. The layout of Table 2 should be fixed.

---

> ### Author Response · Authors · 2022-11-10
> **Response**
>
> Thanks for pointing out that our problem is well-motivated.
>
> ### About bootstrapping and novelty
> According to our understanding, bootstrapping methods for KG construction should be iteratively updating the entity pair set and the pattern (“prompt” in our context) set. However, our framework actually doesn’t consist of such iterations. As shown in Figure 1, we search for entity pairs after we have fixed the prompt set.
>
> There is an iteration in the prompt creation stage (the left box in Figure 1), but we still don’t need to worry about the semantic drift, because all the prompts will be evaluated and weighted with compatibility score to seed entity pairs (Equation 1). As long as there are enough reasonable prompts created in this iteration, which is true as shown by the prompt examples in the appendix, we will get desired prompts and filter out drifted ones for the next stages: knowledge searching and re-scoring. In a word, our design of compatibility score makes sure that the errors will not be propagated to the following stages.
>
> Regarding the novelty and contribution, we want to clarify that our work does have significant contributions to solving a novel knowledge harvest task and tackling technical challenges. Please refer to our general response for our clarification.
>
>
> ### About edit distance
> * Motivation: Previous papers [1, 2] have shown that differently wording but semantically same prompts may lead the LM to generate different contents. In essence we want to get the prompts that are semantically the same as the initial prompt, so it would be running in the opposite direction if we restrict the embedding distance (i.e. with neural methods like SentenceBERT) to be large. Instead, we encourage larger edit distance in order to generate diverse wordings of a relation, which is helpful to harvest more consistent entity pairs. Specifically, we observe the restriction over ED prevents trivial paraphrases like simply adding a “the” to another prompt.
>
> * Details: Each time we get a new prompt, we calculate its ED to all prompts in the current prompt set and check if the maximum of them is lower than a threshold. We only take prompts satisfying this criterion into our prompt set. In the end, every pair of prompts should have ED lower than the threshold.
>
> ### About evaluation
> Regarding the questions on our evaluation, we would like to emphasize that, since we propose a novel task that has not been explored by previous papers, the evaluation is not a trivial job. Human evaluation is the only direct evaluation in our work.
>
> We conduct human evaluation to check the quality of the generated knowledge graphs in two settings: How does the diversity, novelty and precision change as the KG scales (Table 2), how do different prompt creation methods (Table 3) and different language models (Table 4) influence the quality of top predictions. Besides human evaluation, which is the only direct way to evaluate our framework, we also inspect the scoring module as a discriminative model on existing knowledge from ConceptNet (Figure 3) and LAMA (Figure 4).
>
> It’s worth noticing that there are not strict baselines to our work, as we are the first to extract knowledge graphs of arbitrary relations from LMs. However, to provide readers with more contexts of previous research, we list the statistics of different methods, and adapt previous prompt creation methods to our problems making the comparison.
>
> ### About KG completion dataset
> We argue that we already did that in the paper by following a thread of work on KGC [3], and even enlarged their conceptnet dataset and supplemented the LAMA dataset to make our results more solid and generalizable to factual knowledge. The results are shown in Figure 3 and Figure 4, indicating our framework is better than previous prompt creation methods in discriminating positive and negative knowledge tuples from both ConceptNet and LAMA dataset.
>
> ### Technical details and writing
> Thanks for pointing out these problems. We added more details in Appendix 5. Regarding your confusion about the compatibility score, we created a figure to illustrate the process and also included it in the appendix.
>
> [1] Measuring and improving consistency in pretrained language models, TACL 2021 \
> [2] P-adapters: Robustly extracting factual information from language models with diverse prompts, ICLR 2021 \
> [3] Commonsense Knowledge Base Completion, ACL 2016

---

### Official Review · Reviewer_XQg6 · 2022-10-25

**Confidence:** 4
**Correctness:** 3
**Technical Novelty And Significance:** 3
**Empirical Novelty And Significance:** 3
**Recommendation:** 5

**Clarity, Quality, Novelty And Reproducibility:**

Overall, the paper is clear, but the writing could be improved. The authors promise to release the code as well as the resulting KG.

**Strength And Weaknesses:**

Pros:

* The work's most significant strength is its practical pipeline for automatically constructing knowledge graphs (KGs) from pre-trained language models (LMs). Previous research on "LMs as KGs" or "knowledge probing from LMs" has primarily focused on prompt generation and does not cover the entire KG construction pipeline.

Cons:

* One potential shortcoming of the work is that its technical contribution may not be sufficient for a research paper.

* The paper also has a few minor flaws, and the writing could be improved, as listed below:

Minor issues:

* The multi-token issue in entity names. The authors state in Sect. 3.2 that "Each entity consists of one or more tokens". How to resolve the multi-token issue in the likelihood computation? Or, for example, how to compute P_{LM}(t|p,h) if the name of entity t has more than one tokens? In Appx. A.5, the authors state that "we enumerate the number of masks for every entity". Is there a maximum value for such a number? Is it a hyper-parameter?

* Sect. 3.2 says that "The use of the minimum individual log-likelihoods allows us to apply rich thresholding and pruning strategies (e.g., thresholding on log PLM(h|p) for proposing the head entities), as described more in the appendix." However, it seems that there is no related content in the appendix. Or, is it Appx. A.5?

* It would be preferable to include a subsection in Section 4.2 that simply introduces the baselines used in the experiment.

* In Sect. 4.2, the authors state that "we restrict every entity to appear no more than 10 times to improve the diversity of generated knowledge". This, in my opinion, is not reasonable because it would result in a sparse KG. Table 2 also demonstrates this problem.

Typos:

* "Those systems, however, often involves a complex set of components" -> Those systems, however, often involve a complex set of components.



**Summary Of The Paper:**

By probing pre-trained language models with prompts, the paper presents a novel and practical framework for automatically constructing knowledge graphs. The framework's input consists of relation definitions and a small set of seed entity pairs. For each relation, the framework generates and then paraphrases prompts. The extracted entity pairs of each relation are then evaluated using the proposed consistency score. Finally, it reranks these pairs and outputs the top-K results. Experiments show that the resulting KG is diverse and accurate. The proposed automatic prompt generation method is also effective. The resulting KG can serve as a symbolic interpretation
of pre-trained language models.

**Summary Of The Review:**

Overall, I think the paper has both advantages (a practical KG construction pipeline) and disadvantages (some small issues). It can be further improved.

---

> ### Author Response · Authors · 2022-11-10
> **Response**
>
> Thank you for your appreciation that we presents a novel and practical framework for constructing KGs beyond traditional “LM as KGs” or knowledge probing.
>
> *About our contribution*, we want to clarify that our work does have significant contributions to solve a novel knowledge harvest task and tackle technical challenges. Please refer to our general response for our clarification.
>
> *Regarding the repetition restriction*, we applied it because we found that when it’s not imposed, the top predictions sometimes involve the same entity for a lot of times, like a large number of tuples like (man, CapableOf, smell), (man, CapableOf, catch), etc. We were worried that it would give the false impression of high accuracy if all the top predictions are about a few popular and easy-to-predict entities. (in Table 3 & 4 and Figure 2). Therefore, we applied this restriction, and with it, we find our framework able to produce diverse yet accurate predictions. We agree with your idea that eliminating this restriction would result in a dense KG and help explore the limits of scalability of our framework. We will include this result in the future version of this paper. Thanks for your suggestion!
>
> Thanks for raising questions about the writings! We have added more details about pruning in Appendix 5. We also fixed other writing issues you mentioned in our revised paper.

---

### Author Response · Authors · 2022-11-10
**General Response**

We thank all reviewers for their insightful comments. We are encouraged by the reviewers’ appreciation that our problem is well-motivated (pDgC), our framework is novel, practical, and beyond previous research on “LM as KGs” or knowledge probing (XQg6), our automatic prompt generation is effective (XQg6), our paper is well-written and our idea is clearly illustrated (dt4V), and our results are promising (dt4V).

We appreciate reviewers’ valuable suggestions to improve the writing, and **have slightly updated the paper and highlighted the updates in red**. We will respond to several general questions raised by the reviewers here, and will address specific questions in our responses to individual reviewers, respectively.



### About Contributions
We emphasize that our main contribution is that we propose an automatic framework to **extract a KG with arbitrary relations** in an efficient and scalable way, and we apply this framework to harvest KGs from a wide range of popular LMs, indicating LMs alone can be good resources for KG construction, showing a new style beyond traditional work. Besides, our framework **provides a fully symbolic interpretation to the LM**, leading to new insights into knowledge capability of LMs.

**Technically**, we build a pipeline consisting of prompt creation, knowledge searching and rescoring, and we propose the automatic creation of diverse weighted prompts (Sec. 3.1) and the efficient searching for consistency knowledge tuples (Sec. 3.2) to tackle the technical problems of the inconsistency in knowledge extraction and the large searching space of entities.

---

### Decision · Program_Chairs · 2023-01-20

**Decision:**

Reject

**Justification For Why Not Higher Score:**

See meta-review

**Justification For Why Not Lower Score:**

N/A

**Metareview: Summary, Strengths And Weaknesses:**

The authors propose a method to extract knowledge from language models such as BERT. Their approach establishes a set of paraphrase prompts representing a relation. Once this set of prompts is found, they search the space of entity pairs that fit this prompt to relational tuples representing knowledge from these models. This method enables relational knowledge to be extracted from language models at scale.


**Strengths:**
Reviewers all thought that the idea of extracting knowledge graphs from language models was neat.

**Weaknesses:**
In general, the reviewers felt the method lacked conceptual novelty and weren’t convinced by the effectiveness of the proposed method given the low evaluation results. Regarding novelty, I’m inclined to disagree with the reviewers as I do think there’s novelty to the idea. While the method is not completely unique (as the work of West et al., 2022 also looks at few-shot knowledge generation from LMs), it has enough differences.

Regarding evaluations, though, I’m inclined to agree with the sentiment of the reviewers that the evaluations fall short of convincing of the value of the task. While the authors respond to this criticism that their method achieved better results than the baselines they set up, these methods are designed for different task instantiations of the “generate knowledge from LMs" problem. Consequently, even though the BertNet method improves over these methods, it’s not clear they validate the task instantiation in the first place for automatic KG construction (as opposed to other methods for automatic knowledge construction, such as retrieval). I think the authors need evaluations that validate their task instantiation, rather than assuming the premise that KGs should be generated from LMs the "BertNet" way the first place.


**Summary Of Ac-Reviewer Meeting:**

N/A